

# PODE: privacy-enhanced distributed federated learning approach for origin-destination estimation

Sidra Abbas[1], Gabriel Avelino Sampedro[2,3], Ahmad Almadhor[4], Mideth Abisado[5], Mehrez Marzougui[6], Tai-hoon Kim[7] and Areej Alasiry[6]

[1] Department of Computer Science, COMSATS Institute of Information Technology, Islamabad, Pakistan
[2] Faculty of Information and Communication Studies, University of the Philippines Open University, Los Baños, Philippines
[3] Gokongwei College of Engineering, De La Salle University, Manila, Philippines
[4] Department of Computer Engineering and Networks, College of Computer and Information Sciences, Jouf University, Sakaka, Saudi Arabia
[5] College of Computing and Information Technologies, National University, Manila, Philippines
[6] College of Computer Science, King Khalid University, Abha, Saudi Arabia
[7] School of Electrical and Computer Engineering, Yeosu Campus, Chonnam National University, Jeollanam-do, Republic of Korea

Corresponding authors
Sidra Abbas, sidraabbas@ieee.org
Tai-hoon Kim,
taihoonn@chonnam.ac.kr

## ABSTRACT

The statewide consumer transportation demand model analyzes consumers' transportation needs and preferences within a particular state. It involves collecting and analyzing data on travel behavior, such as trip purpose, mode choice, and travel patterns, and using this information to create models that predict future travel demand. Naturalistic research, crash databases, and driving simulations have all contributed to our knowledge of how modifications to vehicle design affect road safety. This study proposes an approach named PODE that utilizes federated learning (FL) to train the deep neural network to predict the truck destination state, and in the context of origin-destination (OD) estimation, sensitive individual location information is preserved as the model is trained locally on each device. FL allows the training of our DL model across decentralized devices or servers without exchanging raw data. The primary components of this study are a customized deep neural network based on federated learning, with two clients and a server, and the key preprocessing procedures. We reduce the number of target labels from 51 to 11 for efficient learning. The proposed methodology employs two clients and one-server architecture, where the two clients train their local models using their respective data and send the model updates to the server. The server aggregates the updates and returns the global model to the clients. This architecture helps reduce the server's computational burden and allows for distributed training. Results reveal that the PODE achieves an accuracy of 93.20% on the server side.

## INTRODUCTION

Statewide consumer transportation demand modeling generally involves several steps, including consumer data collection, model development, calibration, validation, and scenario analysis (*Smith & Fijalkowski, 2022*). Data collection may involve surveys, focus groups, or other methods to gather information on travel behavior and preferences. Model development involves creating mathematical equations that capture the relationships between various factors influencing travel behavior, such as household income, vehicle ownership, and land use patterns (*Brownell & Kornhauser, 2014*). Calibration and validation involve adjusting and testing the models to reflect observed travel behavior accurately. The amount of freight movement has increased during the last few decades. Numerous variables, including population increase, infrastructure development, and the removal of trade barriers, among others, help to explain this (*Tavasszy & De Jong, 2013*). Transportation expenses contribute significantly to supply chain costs, making up 50% of logistics costs and, depending on the industry, around 10% of the entire cost of a good (*Rodrigue, 2020*). Due to technological advancements and the usage of e-commerce, the volume of freight moved at the final mile has dramatically increased in recent years (*Heuser, 2006*; *Sheth et al., 2019*).

It is common practice to employ statewide models, which include passenger and freight movements, to support various statewide planning initiatives. Many governments use them for various planning requirements, including traffic impact assessments, air quality conformance analyses, freight planning, economic development studies, project prioritization, and more (*Horowitz, 2006*). Federal Highway Administration (FHWA) and United States (US) Census Bureau data show that the United States transportation system shipped 17.6 billion tonnes of products in 2011 to support 7.4 million commercial establishments and nearly 117 million households. Trucks are the most common form of freight transportation; in 2011, the industry moved 11.9 billion tonnes, or around 67%, of all the freight moved in the United States (*US Department of Transportation, 2013*). Due to its significant impact on the economy of the state and the country, truck demand has gained relevance in the statewide planning process.

As long as the US economy maintains its steady expansion, international merchandise trade rises, the productivity of the freight sector develops, and there is a need for a large multimodal transport system and truck transport will continue to expand during the next 10 years (*Ammah-Tagoe, 2004*). Based on the Freight Analysis Framework (FAF) database, 75% of all domestic freight shipments are transported by trucks, which will not change between 2007 and 2040 (*US Department of Transportation, 2024*). Meanwhile, the capacity for moving freight, particularly on roads, is growing too slowly to meet demand. This growth mismatch may substantially impact traffic at highway junctions, interchanges, and roadblocks or checkpoints. The inability of trucks to maneuver in congested metropolitan areas because of restrictions on height, length, breadth, weight, incidental loading, or construction, for example, contributes to congestion. The triptable of truck origin–destination (O-D) is utilized by government organizations, service providers, and planners of strategic transportation to pinpoint possible bottlenecks in their regions.

The outcomes of a truck trip table derived from the suggested framework would help state DOTs and metropolitan planning organizations (MPOs) evaluate operational methods to address the ensuing implications of truck traffic, such as traffic problems, congestion, security, and atmosphere. The long-term strategy for infrastructure management could be strengthened by demand prediction.

Many small states typically need more funding to carry out freight surveys or hire technical personnel to create the freight demand model. Furthermore, business freight databases are frequently private and inaccessible to the public. As a result, many models now in use either ignore this factor or erroneously assume that the behavior of freight journeys is the same as that of passenger excursions (*Ogden, 1992*). This might be a possible weakness of truck demand modeling in the statewide model, wherein truck flow properties have been defined by other contributing elements such as location factors (*i.e.,* sites of production and market), physical factors (*i.e.,* how commodities can be transferred: in mass, container, or bulk), geographical conditions, and so on *de Dios Ortúzar & Willumsen (2012)*.

**Motivation:** The decision to adopt federated learning (FL) is driven by the nature of the problem and the desire to use dispersed computing capacity effectively. FL allows the training of our DL model across decentralized devices or servers without exchanging raw data. In the context of OD estimation, sensitive location information of individuals is preserved as the model is trained locally on each device. Only the model updates (gradients) are shared, enhancing privacy and addressing concerns about sharing personal location data. In this scenario, the FL structure consists of two clients and a server. Local models at each client train deep neural networks separately on their portions of the dataset. The knowledge obtained from these local models is then combined into the global model, which is stored on the server. This decentralized training paradigm addresses data privacy and security concerns and enhances the learning process by distributing computing over multiple nodes. The present research aims to pioneer a fresh approach to truck destination estimation using a federated learning-based scheme. This research focuses on creating and deploying a distributed learning methodology where numerous clients can collaborate to train a shared model. This new strategy improves the accuracy of destination predictions and tackles the essential privacy issue by allowing each customer to keep their data hidden. The present research seeks to considerably advance efficient and secure truck destination estimates by fostering collaboration in a federated learning framework, paving the way for improved real-world applications in logistics and transportation systems.

FL fundamentally resolves problems with data silos and privacy. In federated learning, it is feasible to incorporate non-shared data from other hospitals, expanding the sample size and improving the model's precision. Federated learning depends on distributed data sets. Transferring raw data locally is necessary to create a shared model across several devices. This approach ensures the security of the patient. We use federated learning to create an ML model based on remote datasets while respecting the privacy of the data. In a federated machine learning (FML) system, each client (a mobile device, server, or IoT device) has its own dataset and ML model. A centralized global server collects the model parameters of local clients in a federated setting using a centralized main ML (global model) model.

Local client models are trained on a dataset, exchanging updates with the overall ML model (primary model). Without transferring raw data, the primary model iterates through several rounds to get model changes from the dispersed clients. The following justifies the usage of FML:

- The study proposes an approach named *PODE* that utilizes FL to train the deep neural network to predict the truck destination state. The proposed distributed learning approach allows multiple clients to collaborate and train a shared model while keeping their data private.
- The proposed methodology employs two clients and one-server architecture, where the two clients train their local models using their respective data and send the model updates to the server. The server aggregates the updates and returns the global model to the clients. This architecture helps reduce the server's computational burden and allows for distributed training.
- The study uses a variety of evaluation metrics, including accuracy, precision, recall, F1-score, the area under the curve (AUC), and CF, to evaluate the performance of the proposed methodology. Results reveal that the *PODE* achieves an accuracy of 93.20% after the server aggregates data from all clients.

The structure of the study is as follows: 'Literature Review' summarizes the earlier research for Freight Analysis. This research also analyses the FML technique for Freight Analysis, as shown in 'Proposed Approach'. The results of this investigation are presented under 'Experimental Analysis and Results'. The conclusions and prospects for further study are covered in 'Conclusion'.

## LITERATURE REVIEW

We concentrated on many areas of the literature, which serves as the backdrop for this study: a discussion on various models for generating freight, variables and modeling techniques, and studies that highlighted the geographical dependencies in generating freight.

### Freight generation models

Freight generation (FG) models are frequently used to forecast the future amount and value of freight generated in an area. Researchers have investigated various modeling techniques to forecast freight movements at various levels (aggregate and disaggregate). Disaggregate models often concentrate on behavioral elements (such as shippers' behavior in mode choice), whereas aggregate models focus on commodities flows at the regional scale. Different approaches to modeling freight flows have been developed, including trip-based, industry-specific, and commodity-based models. The transported freight is classified as a variety of commodities or commodity groupings in commodity-based models. Estimates of freight movements are made in terms of weight or volume, and later, they are translated to vehicle trips (*Teye & Hensher, 2021*; *Wisetjindawat & Sano, 2003*). Freight flows are calculated in industry-specific models using input and output interactions between common industrial sectors of the economy, according to these cited

researches (*Oliveira-Neto, Chin & Hwang, 2012*; *Ruan & Lin, 2010*; *Sorratini & Smith Jr, 2000*).

The total number of vehicle journeys is approximated in trip-based models as per these cited researches (*Eriksson, 1970*; *Wisetjindawat et al., 2007*; *Gonzalez-Feliu & Sánchez-Díaz, 2019*). These models are referred to in the literature as FG models, which deal with the production of commodities, and freight trip generation (FTG) models, which deal with the movement of freight by vehicles (*Günay, Ergün & Gökaşar, 2016*; *Holguín-Veras et al., 2013*). The overall goal of the current study is to examine FTG.

## Modelling approaches and variables

Various methods have been employed to estimate FG. Time-series analysis, artificial neural networks (ANN), machine learning (ML), economic input–output models and regression are common methods (*Cascetta et al., 2013*; *Holguin-Veras, López-Genao & Salam, 2002*; *Hassan, Mahmassani & Chen, 2020*). Economic demand and transportation supply are made up of the freight transportation system. Due to its connection to the manufacturing and distribution of tangible items, freight transportation is linked to economic activity in an area or nation. Regression and economic input–output models link socioeconomic data with freight activities, two common econometric techniques used in aggregate modeling (*Pendyala, Shankar & McCullough, 2000*). Various factors influence FG; population indices, employment, several enterprises, zonal area, industrial floor space, and land use are frequently employed as explanatory variables for the estimate. Numerous studies have revealed the importance of population and employment, with the former being critical to output and freight attractiveness (*Bastida & Holguín-Veras, 2009*; *Iding, Meester & Tavasszy, 2002*). According to *Brogan, Brich & Demetsky (2002)*, fifteen important commodities were selected for Virginia, and regression models for freight production and attractiveness were created. As predictor factors, characteristics related to particular commodities, population, employment, county/city size, per capita income, and population density were included. With employment, location, and commercial sector as the explanatory variables, *Sánchez-Díaz (2017)* proposed a framework and constructed FG models for commercial premises in Gothenburg. The research demonstrated that socioeconomic factors effectively predict freight demand.

## Spatial interactions in FG

Few researchers have specifically addressed geographical and locational impacts in freight transportation, although several have documented their significance (*Bhat & Zhao, 2002*; *Cohen, 2010*; *Miller, 1999*). Recent research by authors, which examined the geographical challenges connected with national-level FG models, underlined the importance of spatial interconnections (*Novak et al., 2011*; *Sánchez-Díaz, Holguín-Veras & Wang, 2016*; *Krisztin, 2018*). The geographical problems connected to national-level FG models were investigated by *Novak et al. (2011)*. To account for global (all the zones are correlated) and local (only nearby zones are associated) spatial spillovers, they created a spatial autoregressive model and a spatial moving average model. At the urban level, *Sánchez-Díaz, Holguín-Veras & Wang (2016)* investigated spatial regression models for freight trip attraction. They looked

at considering spatial interconnections using network and land use characteristics. *Krisztin (2018)* used a semi-parametric spatial autoregressive method to consider both spatial dependence and non-linearities in the model parameters. In the model, to account for geographical dependency and non-linearities and analyze the growth patterns of 155 European areas, *Basile (2008)* created a semi-parametric spatial Durbin model. To acquire the interactions between the commodities using a path analysis technique considering socio-demographic, land use, and firm-related factors, *Hensher & Teye (2019)* built a commodity generation chain model at a national level. When working with cross-sectional data, several economists have emphasized the importance of spatial interaction (*LeSage & Pace, 2009*; *Anselin & Bera, 1998*). Biased estimates may emerge from disregarding the geographical dependence of the dependent variables, and inefficient estimates may result from disregarding the spatial dependence of the disturbance factors. Therefore, while modeling regional FG, it is crucial to consider geographical interdependence and spatial autocorrelation.

## PROPOSED APPROACH

This section describes the technique used for this research, including major stages shown in Fig. 1. This method focuses on using the NHTS Truck OD dataset for classification tasks. Notably, three critical preparation stages—resolving missing values, data normalization, and dimensionality reduction—were rigorously conducted to refine the dataset before experimentation.

Our methodology is based on federated learning, which involves training a deep neural network with two clients and a central server. The two clients' localized training leads to a global model aggregating their findings, allowing for a decentralized yet collaborative learning process. To improve the interpretability and performance of our analysis, we pared down the original 51 target labels to a more manageable collection of 11. During the model construction phase, a carefully balanced distribution of the dataset was used for testing, validation, and training, with 30% set aside for testing, 20% for validation, and 50% for training. The use of deep learning techniques during the training stage proved beneficial in providing the model with the capacity to detect detailed patterns within the dataset. It is worth noting that the dataset under consideration contains a considerable 209,866 rows, which contributes to a strong and thorough training, validation, and testing framework. This large dataset increases the reliability and generalizability of our findings. Overall, thorough preprocessing, federated learning, and smart dataset dissemination constitute the foundation of our research technique, allowing for insightful and meaningful results.

### Dataset selection

As part of a larger effort to gather and analyze data on travel behavior for passenger and freight transportation in the United States, Oak Ridge National Laboratory (ORNL) has created this dataset. The NHTS Truck OD dataset details trucking activity in the United States, including trip origins and destinations, modes of transportation, and other trip parameters like distance, length, and purpose. The dataset includes information from both personal and commercial trucks and spans the entirety of 2020. The 2020 national truck

_Peer_J Computer Science ______________________________________________

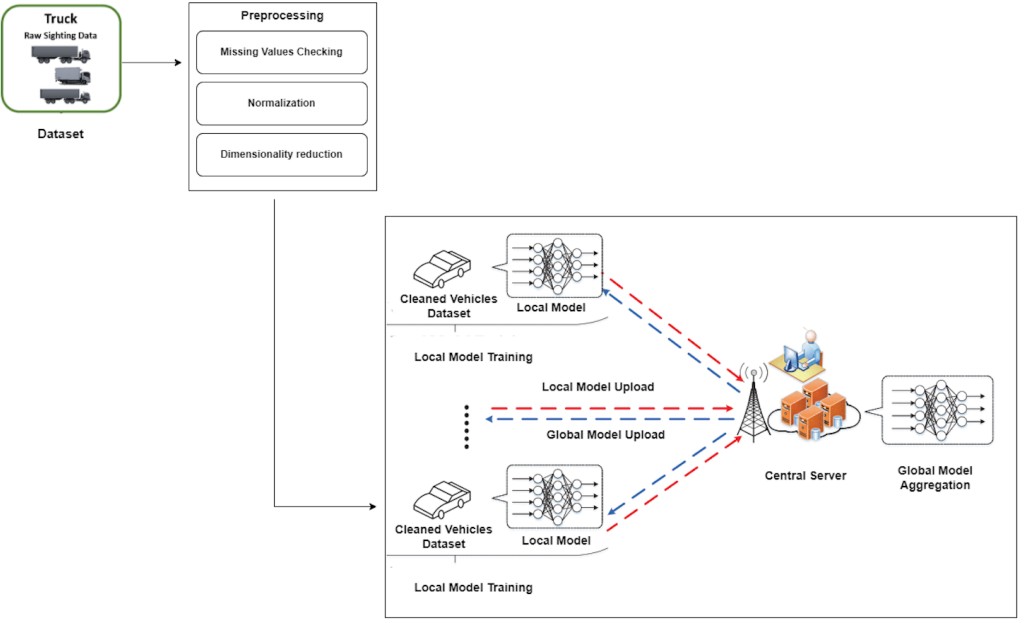

**Figure 1** **Proposed approach for truck data classification using federated deep neural network.**

data provide insights into population-level truck travel demand by trip lengths. All 50 states and DC are represented in the OD tables, which aggregate data from 583 MSAs and non-MSA zones. There are 339,889 rows in the datasets, along with 19 features and 51 labels. Most of the information in the dataset is missing, and we removed those columns.

## Data pre-processing

Preprocessing is important because it prepares the data for analysis by removing missing data and inconsistencies. It ensures that the data is in the correct format, comprehensive, and ready for the algorithm's training. The preparation process includes common strategies used to clean and manipulate data. Delete columns with null values to decrease noise in the data and simplify spotting trends. By deleting null values from rows, you ensure that the dataset is full and that there are no missing values that could lead to bias in the study. Filling in missing values with linear interpolation is an excellent approach to impute missing values without biasing the data. This method generates new values based on the existing values in the dataset, which aids with data integrity. Dropping columns with null values reduces the dataset's dimensionality, making it easier to analyze and visualize. Lowering the complexity of the model can also improve the machine learning algorithm's performance. Initially, Fig. 2 shows the target columns presented in the original dataset. Firstly, this research removed the missing values according to the target column. This yields a total of 20,9866 rows and 19 columns, but the dataset still has missing values. Further, this research handles the remaining values that need to be added. We require a more suitable approach that will maintain the majority of data. To create a new value using the linear interpolation method, use the interpolation method. We must find a way

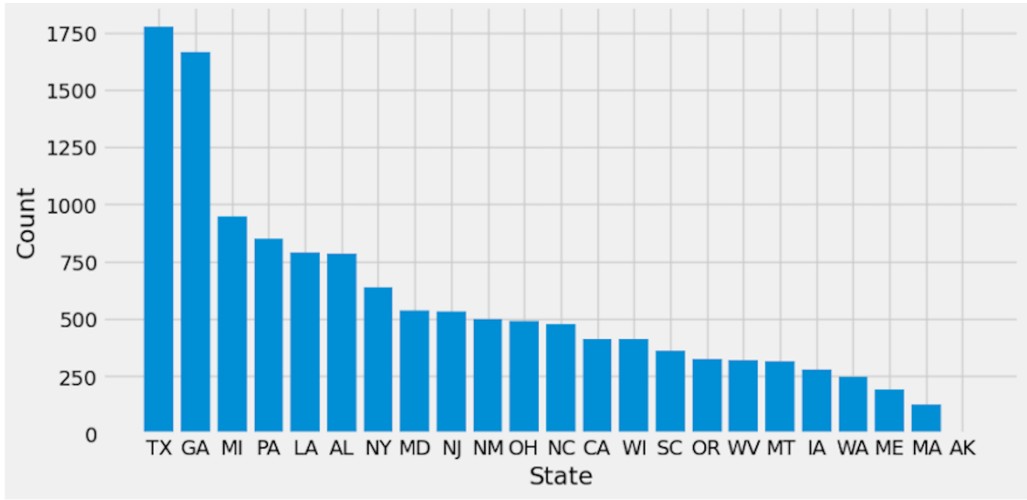

**Figure 2** Original dataset labels.

to substitute the mean when dealing with many missing variables. Doing so would result in many identical (duplicate) values, throwing off the AI system. That is why we removed the remaining missing values. Finally, we have 209,866 rows and 15 columns. The dataset contains 51 target labels, and all of them are not necessary. This research set the threshold at 7 k frequency to lower the total number of classes. Remove any categories that occur less than 7,000 times. Finally, we obtained 11 unique and important target labels as shown in Fig. 3. The pre-processing methods included filtering away categories that appeared less than 7,000 times to improve the learning process. By lowering the number of target labels from 51 to 11, we aimed to alleviate the difficulties associated with many classes. This reduction speeds the learning process and tackles potential concerns with imbalanced class distribution, which can negatively impact model training.

While this reduction may appear to reduce the task's difficulty, our major goal was to improve the model's capacity to generalize and produce meaningful predictions in the most relevant and frequent categories. This technique is consistent with best practices in machine learning, where a balance of job complexity and model performance is frequently sought.

To be more clear, we will explicitly state in the work that reducing the number of target labels was intended to optimize the learning process by focusing on the most relevant and prominent categories, potentially enhancing the model's overall performance. We appreciate the reviewer's feedback and will integrate this explanation into the updated text.

Furthermore, this research normalized the data using *MinMaxScaler*. *MinMaxScaler* is a data normalization technique that is used in data preprocessing. It scales and modifies the dataset's features to fall within a given range, often between 0 and 1. This is useful for algorithms like gradient descent that are sensitive to the scale of the input characteristics.

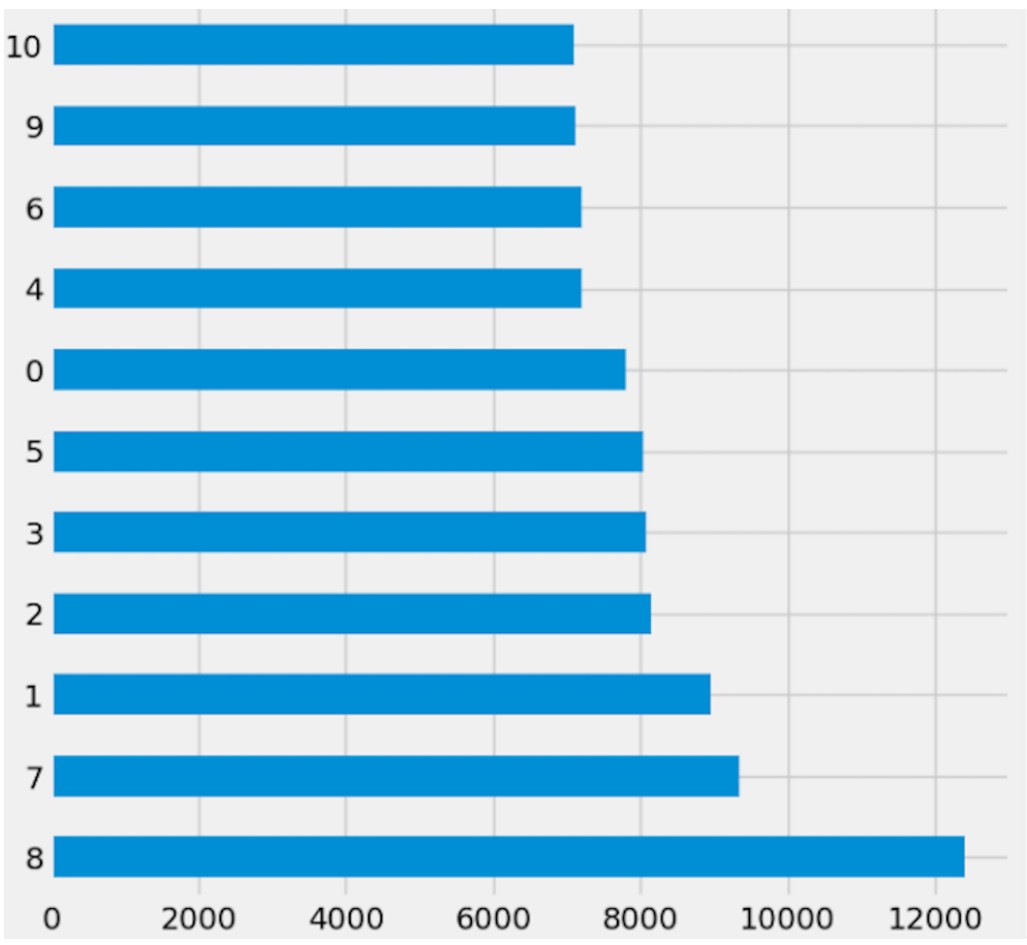

**Figure 3    Reduced dataset target labels.**

The MinMaxScaler transformation formula iS is shown in Eq. (1):

$$X_{scaled} = \frac{X - X_{min}}{X_{max} - X_{min}} \tag{1}$$

where 'X' is the original feature value. 'X_min' is the minimum value of the feature in the dataset. 'X_max' is the maximum value of the feature in the dataset. 'X_scaled' is the transformed feature value scaled between 0 and 1.

## Framework of the federated deep learning

This section explains how federated deep learning works and its basic structure for truck data classification. The FL process involves two clients and a server aggregating each client's results to get an overall result. The server maintains the global model and sends various parameters to different consumer nodes while each client is trained separately on its local device to protect its privacy. The server extracts information from each client, updates the global model, and sets the score. This approach ensures that the client terminal is accurate and private and that the computing power of the client terminal and consumer data is used to learn and keep a good global model. Clients provide accurate data and use their private

data to train the model, submitting it to the FL server. The data owner (client) participates in the FL process and collaborates with others to develop the aggregate server center's DL model. FL training consists of three stages: task initialization, local model training and updates, and global model aggregation and updating. In the first step, the server determines the training task and sets parameters like the learning rate. In the second step, each client updates the local model parameters using local data and equipment to minimize the loss function. In the third step, local user models are aggregated, and updated main model parameters are sent to data owners *via* the server.

The goal is to minimize the loss function for each participant. Finally, in the global model aggregation and updating stage, the local models (client updates) are aggregated, and the updated main model parameters are sent to data owners *via* the server. In the global model aggregation and updating stage, individual client updates to local models are aggregated. These changed major model parameters are merged before the enhanced model is distributed to data owners *via* the server. By defining this aggregation rule, we hope to increase the transparency of our research methods by providing a clear knowledge of how information is combined and model parameters are modified during the global aggregate process.

## Classification model

This study used a deep neural network (DNN) model implemented in Python using the Keras library for federated learning with two clients and one server. The model classifies truck-related data using the training data features.

One input layer, six hidden layers, and one output layer make up the model's architecture. ReLU activation function, a popular activation function used in deep learning models, is present in 512 neurons in the input layer. The ReLU activation function is present in the input and hidden layer neurons, respectively, across the six hidden layers. Last but not least, 11 neurons in the output layer have the softmax activation function, which is frequently employed in multi-class classification problems.

The "sparse_categorical_crossentropy" loss function, "adam" optimizer, and "accuracy" metric are used in the model's construction. For multi-class classification issues where the classes are mutually exclusive, the "sparse_categorical_crossentropy" loss function is employed. The "adam" optimizer is an algorithm commonly used in deep learning models. With a validation split of 0.2 (20% of the data is used for validation), the model is trained on the data using the "fit" function for 15 iterations. The "hist" variable tracks and stores the accuracy and loss metrics throughout training. Each client's data is used to train the model separately, and the server aggregates the weights to update the overall model. Finally, the model architecture is made to identify useful features in the truck data and use deep learning to categorize it into one of the 11 classes.

## Proposed approach algorithm

The proposed approach for predicting truck destination state using federated learning is shown in Algorithm 1. Let the dataset be represented by a matrix X, where each row corresponds to an instance, and each column corresponds to a feature. Suppose there are

missing values in some entries of X, denoted by NaN. We can replace these missing values with the linear interpolation value of the corresponding feature. The linear interpolation method replaces the missing value with the average of the two nearest non-missing values of the same feature. If there is only one non-missing value, the missing value is replaced with that value. MinMaxScaler normalizes each feature to have values between 0 and 1. The normalized value of an entry is represented by $X\_i,j$. where $X\_:,j$ denotes the jth column of X. Next, the preprocessed dataset was divided into two clients for training. Here, DNN represents a deep neural network, $\mathcal{D}_c$ represents the dataset of client $c$, $\alpha$ and $\beta$ represent the learning rates for local and global model updates. The algorithm initializes the global and local model parameters, updates the local model parameters using stochastic gradient descent, aggregates the local model parameters from all clients, updates the global model parameters using the aggregated local model parameters, and evaluates the global model using various metrics (accuracy, precision, recall, F1-score, area under the curve (AUC), and confusion matrix (CM)).

# EXPERIMENTAL ANALYSIS AND RESULTS

This section describes the FDNN model's experimental outcomes and subsequent discussion for rating classification on the Truck dataset. Model testing used 30% of the dataset, validation used 20%, and training used 50%. Deep learning was employed during the model's training process. The 'sparse_categorical_crossentropy' loss function and the 'adam' optimizer were used to create the final model. Accuracy, precision, recall, F1-score, and CM were just a few evaluation measures used to assess the model's performance. This section discusses the experimental findings in detail. Given the unique nature of our dataset, which has not been used in previous studies, direct comparisons with known baselines pose a significant barrier. However, we understand the importance of demonstrating the benefits of our plan through a relevant comparative analysis. In pursuit of this goal, we chose a federated learning technique comprising two clients. The decision to limit the number of clients to two was motivated by resource concerns. Despite these limits, comparing the tests done with two clients provided useful insights, allowing us to successfully showcase the benefits of our suggested method within the constraints of our available resources. This decision illustrates our determination to derive significant results and contribute valuable insights to the topic despite resource constraints.

## Experimental setup

The research used an HP Omen 15 laptop running Windows with an Nvidia 1060 GPU for acceleration and Python 3.8.8 as the programming language. Pycharm was used as the development framework. Pycharm is a Python IDE that streamlines the process of writing, debugging, and testing program code. Windows is a popular OS because it provides a solid platform to deploy Python software. The HP Omen 15 laptop's robust processor and large memory make it a great place to test new machine-learning techniques. Fast and efficient parallel processing from the Nvidia 1060 GPU shortens the time it takes to train and evaluate deep learning models. Python 3.8.8 is a widely used language for machine learning because of its extensive collection of libraries and tools for this purpose.

---

**Algorithm 1** Truck destination state prediction using Federated Learning

---

**Require:** *NHTS Truck OD dataset*

**Ensure:** *Predicted destination state*

1: **Handle missing values,**

2: *Dataset ← X*

3: **for** each feature $j$ in $X$ **do**

4:      **for** each instance $i$ in $X$ **do**

5:          **if** $X_{i,j} = \mathrm{NaN}$ **then**

6:             Let $X_{i,j}$ be the average of the two nearest non-missing values of $X_{i,j}$, using linear interpolation.

7: Output the matrix $X$ with missing values removed.

8: **Normalize the numerical features**, using the min-max scaler: $X'i,j = (Xi,j - \min(X_{:,j}))/(\max(X_{:,j}) - \min(X_{:,j}))$

9: Split dataset into two clients and one server

10: Initialize global model parameters $\theta^0$

11: **for** each client $c \in 1, 2$ **do**

12:      Initialize local model parameters $\theta_c^0$

13:      **for** each round $r \in 1, 2, 3$ **do**

14:          Update local model parameters using client's data:

15:          $\theta_c^r = \mathrm{DNN}(\theta_c^{r-1}, \mathcal{D}c, \alpha)$

16:          Send updated model parameters to the server

17:          Aggregate local model parameters from all clients:

18:          $\hat{\theta}^r = \frac{1}{2}\sum c = 1^2 \theta_c^r$

19:          Update global model parameters using aggregated local model parameters $\theta^r = \mathrm{DNN}(\theta^{r-1}, \hat{\theta}^r, \beta)$

20: Evaluate the global model using precision, recall, F1-score, the area under the curve (AUC), and the CM.

21: $accuracy = (TP + TN)/(TP + TN + FP + FN)$

22: Return the accuracy.

---

## Evaluation metrics

Accuracy, precision, recall, F1-score, the area under the curve (AUC), and CM were the metrics this research used for evaluation. All of these were crucial to the success of the model. These metrics are used to assess the effectiveness of the models in this research. Check the model's performance with the provided metrics. The following methods can be used to access the model's detection capabilities:

**Accuracy:** The model's accuracy is the percentage of times it correctly labels training data. This means it only adds the samples accurately predicted to the comprehensive set of projected samples. Accuracy is mathematically expressed in Eq. (2).

$$Accuracy = \frac{(TP + TN)}{(TP + FP + TN + FN)}. \tag{2}$$

**Precision:** Precision is the ratio of correctly predicted successes to total successes. Eq. (3) presents the mathematical form of precision.

$$Precision = \frac{(TP)}{(TP + FP)}.$$ (3)

**Recall:** The percentage of correctly identified positives is known as recall. The total number of true positives is calculated by subtracting the number of false negatives and genuine positives. Recall is represented mathematically in the Eq. (4).

$$Recall = \frac{(TP)}{(TP + FN)}.$$ (4)

**F1-score:** The F1-Score is the harmonic mean of the model's precision and recall, calculated by adding these two metrics together. It can enhance the evaluation process when datasets are not evenly distributed. The formula for the F1-score is shown in Eq. (5).

$$F1 - score = \frac{(2 * (Precision * Recall))}{(Precision + Recall)}.$$ (5)

**Confusion matrix (CM):** is used to evaluate the accuracy of a model by comparing predicted and actual classifications. However, a unique approach is often used to create a CM that ensures the model is not overfitted to the training data. This is accomplished by dividing the data into two sets: a training set and a validation set. The model is trained on the training set, and the CM is generated based on the predictions made on the validation set. This ensures that the model is not simply memorizing the training data but learning to generalize and perform well on new, unseen data. The model's accuracy can be more accurately evaluated, and the risk of overfitting can be mitigated.

## Server-side training results

In this analysis of the FL system, we focus on the main parameter server and two consumers. The server monitors all clients and receives their model updates while training the deep neural network model. The data logs document the training of the DNN model on the server side. To track down a specific operation, you can consult the server's logs. Each of our experiments is scored three times. We conducted the DNN analysis the first three minutes after the FL system started. There are two stages, fitting and evaluating, in every iteration. During the fitting phase, the client sends training data to the server; during the evaluating phase, both clients send test data to the server. Then, we compile every piece of information. We completed the experiment in 17 min using the given methods. The 93.20% accuracy of the DNN model is achieved when the server collects data from all clients. The outcomes prove that the DNN model performs well on the truck dataset.

## Client 1's experimental results

Truck categorization with a federated deep neural network requires several clients and a server that aggregates each client's results to produce the overall result. The server maintains the global model and transmits different parameters to user nodes. To maintain their privacy, each client is taught about their local data. The server collects input from each client, updates the global model, and assigns a score. The client then uploads the

**Table 1  Client 1's DNN model experimental results.**

| Experiments | Accuracy % | Precision % | Recall % | F1-Score % |
|---|---|---|---|---|
| Round 1 | 81.03 | 81.04 | 81.03 | 81.02 |
| Round 2 | 87.04 | 88.04 | 87.04 | 87.01 |
| Round 3 | 90.03 | 90.04 | 90.04 | 90.03 |

revised model parameters to the server, which changes the model parameters. For each experiment round, the accuracy, precision, recall, and F1-score for client 1 are displayed in Table 1. The accuracy, precision, recall, and F1-score in round 1 were 81.03%, 81.04%, 81.03%, and 81.02%, respectively. The accuracy, precision, recall, and F1-score in round 2 were 87.04%, 88.04%, 87.04%, and 87.01%, respectively. The accuracy, precision, recall, and F1-score in round 3 were 90.03%, 90.04%, 90.04%, and 90.03%, respectively. Each round's results reveal that the accuracy, precision, recall, and F1-score rose. The results show that the model's performance improved with each round of training on Client 1's data, indicating that the federated learning approach is effective for truck classification.

The study also presented the graphs of the CM and the accuracy and loss throughout training and validation. Figure 4 displays the result's visualization of client 1. The results of client 1 in round 3 were the highest compared to those of round 2 and round 1. The CM of round 3 is shown in Fig. 4G. The diagonal of the CM represents the instances that the model correctly classified, while the non-diagonal elements represent instances that were misclassified. Specifically, the element at row *i* and column *j* represents the number of instances that belong to class *i* but were predicted to belong to class *j*. Figures 4H and 4I show the DNN model training, validation accuracy, and loss of round 3. The model training accuracy is 94%, and validation accuracy is 88%, while the model training loss is 0.10 and validation loss is 0.41%.

## Client 2's experimental results

Federated deep neural network (FDNN) for truck classification is a privacy-protecting method of training a model on dispersed client devices. The goal of FDNN is to learn from the information of many clients in a way that protects their anonymity. Table 2 displays the outcomes of the experiments performed for client 2. The data for three training iterations are displayed in the table below. Client 2 performed with an 83.02% F1-score, 83.02% accuracy, 83.02% precision, and 83.02% recall in Round 1. Through iteration two, we increased our accuracy to 89.10%, precision to 89.11%, recall to 89.11%, and F1-score to 89.10%. After three iterations, the accuracy, precision, recall, and F1-score improved to 91.01%. Results for client 2 show steady increases in quality across the board, from accuracy and precision to recall and F1-score. As more customers participated in the training process, the model benefited from exposure to a larger and more varied dataset, leading to better results. Each client's local data was used to refine the model's parameters, resulting in highly accurate training tailored to the individual. Each round's findings show that the FDNN model can efficiently learn from data belonging to numerous customers while keeping their identities secret. Accuracy and loss were plotted during training and validation, as was the CM. The results of client 2 are visualized in Fig. 5. Compared to round

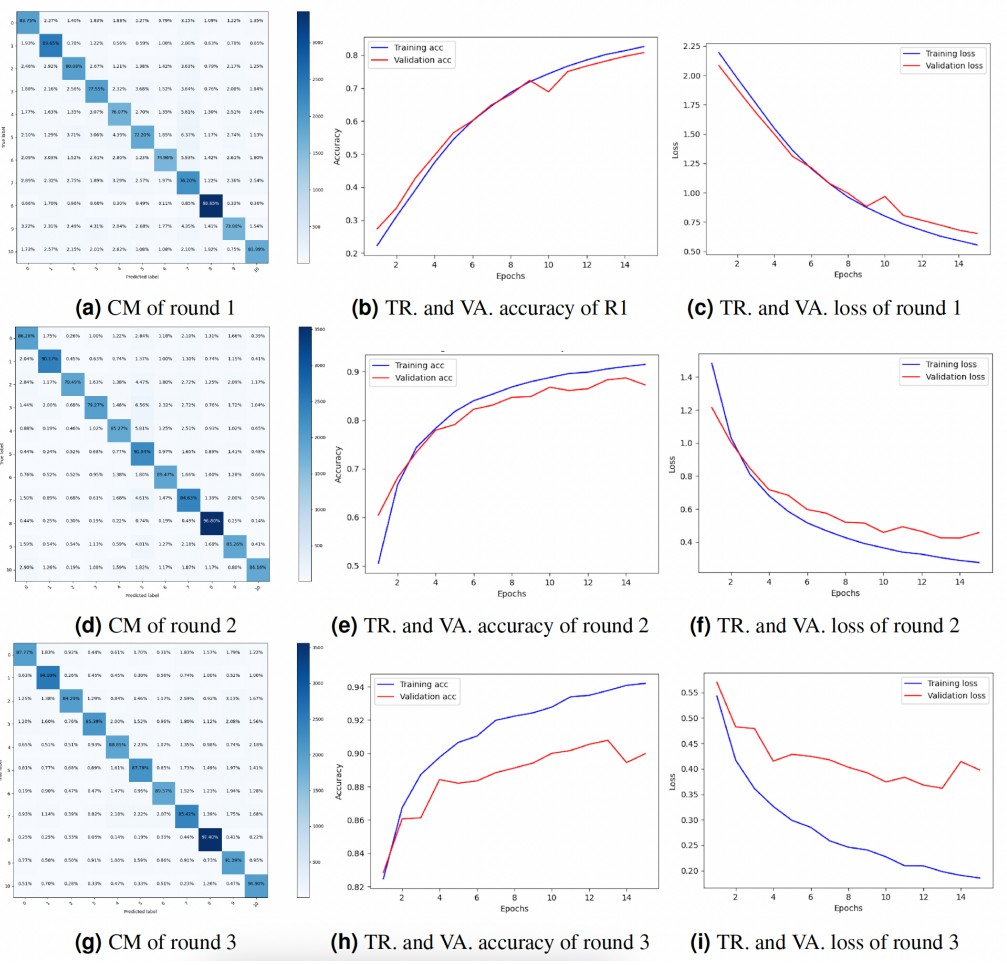

**(a)** CM of round 1    **(b)** TR. and VA. accuracy of R1    **(c)** TR. and VA. loss of round 1

**(d)** CM of round 2    **(e)** TR. and VA. accuracy of round 2    **(f)** TR. and VA. loss of round 2

**(g)** CM of round 3    **(h)** TR. and VA. accuracy of round 3    **(i)** TR. and VA. loss of round 3

**Figure 4** **Graphical representation of client 1 DNN results.** Key: TR, Training; VA, Validation.

**Table 2** **Client 2's DNN experimental results.**

| Experiments | Accuracy % | Precision % | Recall % | F1-Score % |
|---|---|---|---|---|
| Round 1 | 83.02 | 83.02 | 82.02 | 83.02 |
| Round 2 | 89.10 | 89.11 | 89.11 | 89.10 |
| Round 3 | 91.01 | 91.01 | 91.01 | 91.01 |

2 and round 1, client 2's outcomes in round 3 were the best. Figure 5G depicts the CM of client 2 in round 3. The diagonal of the CM represents the cases successfully identified by the model, whereas the non-diagonal entries reflect the misclassified instances. Row $i$ and column $j$ contain the total number of instances misclassified as belonging to class $j$ while belonging to class $i$. The accuracy and loss of the DNN model in round 3 during training and validation are displayed in Figs. 5H and 5I. With a training loss of 0.15 and a validation loss of 0.38, respectively, the model training accuracy is 94%, and the validation accuracy is 91%.

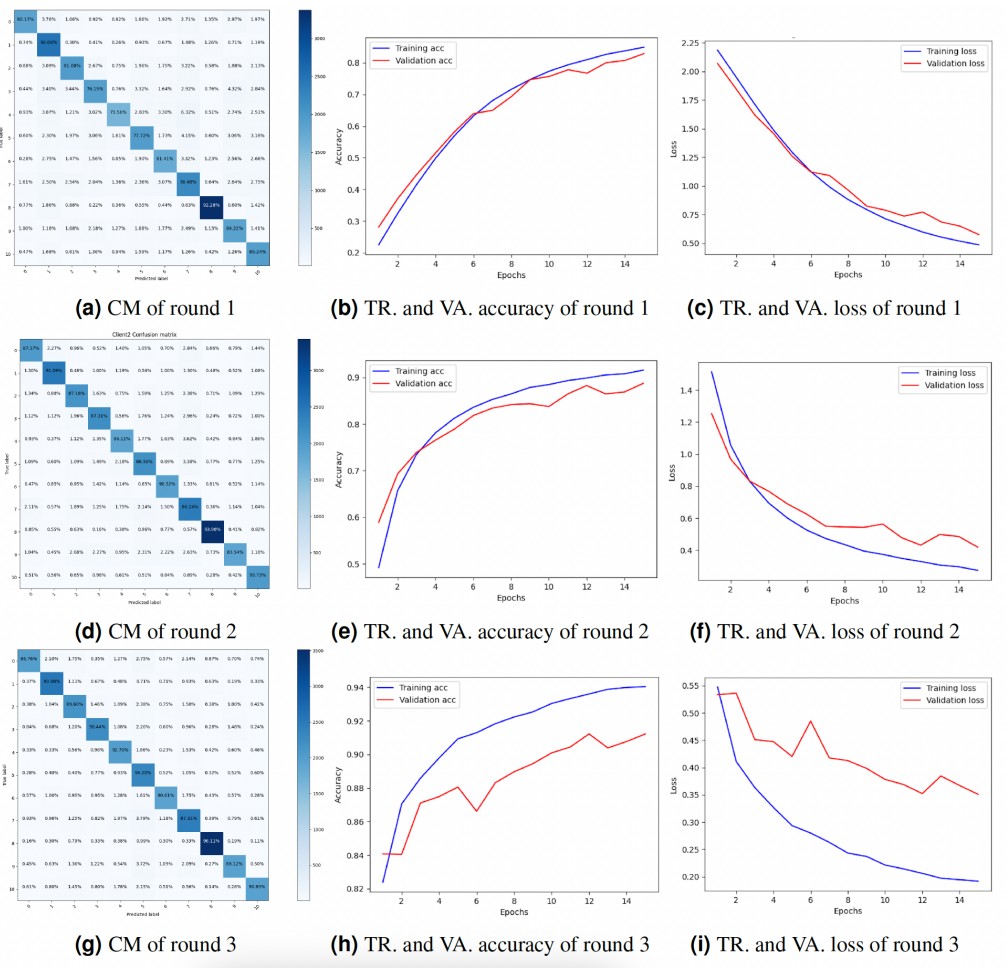

**Figure 5** **Graphical representation of client 2 DNN results.** Key: TR, Training; VA, Validation.

## CONCLUSION

This study has considerably increased our understanding of the complex interplay between vehicle design changes and their impact on road safety, using naturalistic research, crash data, and computerized driving simulations. The increasing integration of computerization in autos emphasizes the need to investigate such paths. The present research makes a unique contribution by investigating a previously unexplored data source for explaining highway fatalities: the National Highway Traffic Safety Administration's database of car owner complaints. Using the NHTS Truck OD dataset, we adjusted our approach by reducing the target label count from 51 to 11, improving the interpretability and efficiency of our analysis. A federated learning deep neural network consisting of two clients and a server is central to our inquiry and critical for processing the NHTS Truck OD dataset. The federated learning paradigm enables decentralized model training, addressing privacy concerns while maximizing the learning process. Combining two local models training deep neural networks yields a global model incorporating their lessons. We completed the experiment

in just 17 min by carefully following the instructions. The DNN model can be as accurate as 93.20% when the server pools client data. Limitations of this work could include using a limited dataset (NHTS Truck OD) with a reduced number of target labels, which may not represent the full range of highway fatalities. Additionally, using car owner complaints as a data source may not fully capture the causes and factors leading to accidents, and there may be biases in reporting such complaints. Future work could incorporate additional data sources, such as police reports or hospital records, to further investigate the relationship between vehicle design changes and road safety. Additionally, expanding the number of target labels used in the analysis could provide a more comprehensive understanding of highway fatalities. Finally, exploring other machine learning algorithms or techniques, such as reinforcement learning or transfer learning, could improve accuracy and performance. This research recognizes the lack of privacy-preserving techniques beyond typical federated learning concepts. It is acknowledged that traditional federated learning has inherent privacy leakage problems. In response to this concern, the research will be amended to include a more thorough assessment of the privacy concerns related to this study's federated learning technique. This includes an examination of the privacy implications and potential risks, as well as a discussion of privacy-preserving approaches that could be used in future iterations of this study. The goal is contributing to a more reliable and privacy-conscious federated learning architecture.

### Funding
This work was supported by the Deanship of Scientific Research at King Khalid University for funding this work through small group Research Project under grant number RGP1/417/44. The funders had no role in study design, data collection and analysis, decision to publish, or preparation of the manuscript.

### Grant Disclosures
The following grant information was disclosed by the authors:
The Deanship of Scientific Research at King Khalid University through small group Research Project: RGP1/417/44.

### Competing Interests
The authors declare there are no competing interests.

### Author Contributions
- Sidra Abbas conceived and designed the experiments, performed the experiments, performed the computation work, prepared figures and/or tables, authored or reviewed drafts of the article, and approved the final draft.
- Gabriel Avelino Sampedro conceived and designed the experiments, performed the experiments, analyzed the data, performed the computation work, prepared figures and/or tables, authored or reviewed drafts of the article, and approved the final draft.

- Ahmad Almadhor conceived and designed the experiments, performed the experiments, performed the computation work, prepared figures and/or tables, authored or reviewed drafts of the article, and approved the final draft.
- Mideth Abisado conceived and designed the experiments, performed the experiments, analyzed the data, performed the computation work, prepared figures and/or tables, authored or reviewed drafts of the article, and approved the final draft.
- Mehrez Marzougui analyzed the data, prepared figures and/or tables, authored or reviewed drafts of the article, and approved the final draft.
- Tai-hoon Kim analyzed the data, prepared figures and/or tables, authored or reviewed drafts of the article, and approved the final draft.
- Areej Alasiry conceived and designed the experiments, prepared figures and/or tables, authored or reviewed drafts of the article, and approved the final draft.

## Data Availability

The 2021 annual truck OD data (including data dictionary) is available at the US Department of Transportation Federal Highway Administration: https://nhts.ornl.gov/od/downloads.

## Supplemental Information

Supplemental information for this article can be found online at http://dx.doi.org/10.7717/peerj-cs.2050#supplemental-information.

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
