# Peer review of "PODE: privacy-enhanced distributed federated learning approach for origin-destination estimation"

_PeerJ Computer Science, doi:10.7717/peerj-cs.2050_

## Round 0.1 · original submission · Major Revisions

Upon reviewing feedback from qualified reviewers, it is evident that your manuscript necessitates significant revisions. Make major changes based on the comments provided. Ensure to address these revisions thoroughly before resubmitting your manuscript.

**Language Note:** The review process has identified that the English language must be improved. PeerJ can provide language editing services - please contact us at [email protected] for pricing (be sure to provide your manuscript number and title). Alternatively, you should make your own arrangements to improve the language quality and provide details in your response letter. – PeerJ Staff

·

Basic reporting

The manuscript conducts a federated learning-based scheme on truck destination estimation. It seems like pre-processing a centralized dataset and then splitting it into 2 subsets to adopt the FL setting.
1. The motivation for using such an FL-based scheme is not highlighted.
2. When doing the pre-processing, it is not clear whether to delete the category numbers, which will decrease the difficulty of such a learning task. It is also not sure whether this way can benefit the learning performance as the authors stated in the manuscript.

Experimental design

1. There is no baseline to compare with the proposed scheme, which cannot show the advantages.
2. The aggregation rule applied to the central model is not clarified.
3. There are no further privacy-enhanced technologies used in the work, where traditional FL has a privacy leakage risk.

Validity of the findings

1. The motivations for applying FL are not clear.
2. It seems like an application on the truck destination estimation scenario, which shows limited technical contributions.

Reviewer 2 ·

Basic reporting

This paper proposed an approach to examine the National Highway Traffic Safety Administration's database of car owner complaints, a rarely examined data source on traffic deaths. The primary components of this study are a customized deep neural network based on federated learning, with two clients and a server, and the key pre-processing procedures.
1. The authors utilised a number of abbreviations in their paper; it would be prudent to define each one prior to its application.
2. Include the description of each section to quickly understand purpose.
3. Try to use past tense in whole paper or may be present indefinite.
4. Why did you select two clients? Is there any good rationale?
4. Sources of the dataset may be specified in detail (the author suggests that only the time could be included, not the dataset's extent). It is stated that data was utilised; please specify whether the sum or mean of the voxel values was utilised.
5. Further elaborate the methodology in the Methodology section and what was the dataset ratio for the proposed approach?
6. Provide a brief theoretical explanation of the machine learning techniques under discussion in Section 3.
7. Construct a more comprehensive conclusion
8. Certain references are incomplete and is hard to follow for readers. Also check mistakes in the communication.

Experimental design

Overall Experimental design is appropriate but still following suggestions need to be addressed in the final version.

Elaborate the methodology in the Methodology section and what was the dataset ratio for the proposed approach?
Construct a more comprehensive conclusion.?
Certain references are incomplete and is hard to follow for readers. Also check mistakes in the communication.?

Validity of the findings

Overall Research validity of this article is appropriate and according to the standard of the journal.

Additional comments

N/A

Reviewer 3 ·

Basic reporting

The paper titled "PODE: Privacy-Enhanced Distributed Federated Learning Approach For Origin-Destination Estimation", seems very interesting research. very impressive. This paper proposes an approach named PODE to examine the National Highway Traffic Safety Administration’s database of car owner complaints, a rarely examined data source on traffic deaths. The primary components of this study are a customized deep neural network based on federated learning, with two clients and a server. This architecture helps reduce the server’s computational burden and allows for distributed training. The paper is clearly written in a good style but need more empirical experiments to prove the objectives. Some latest research work needs to be cited .

Experimental design

The objective and motivation for the research has been very well stated in the introduction part. But needs clarification on the following:

1. The proposed methodology seems very basic and need more detailed specification to synchronize with objectives..
3. Novelty of research work and finding needs more discussion in result and discussion section.

Validity of the findings

The authors adequately evaluated their work, but all claims need more justification and should be supported by empirical experiments.

Additional comments

This research paper is very interesting study. Problem statement is formulated in a very significant way. The research conducted seems very basic. By improving the methodology and enhanced empirical experiments this paper can be improved

---

## Round 0.2 · accepted · Accept

Please follow the next steps from the production team

Reviewer 2 ·

Basic reporting

According to the journals standard.

Experimental design

Appropriate

Validity of the findings

Good

Additional comments

N/A

Reviewer 3 ·

Basic reporting

The paper titled "PODE: Privacy-Enhanced Distributed Federated Learning Approach for Origin-Destination Estimation" has improved by authors as per suggestions. Introduction section has updated by highlighting the motivation for the research.

Experimental design

OK

Validity of the findings

Justified and Satisfactory

Additional comments

not required